# Megakaryocyte emperipolesis mediates membrane transfer from intracytoplasmic neutrophils to platelets

Pierre Cunin[1], Rim Bouslama[1], Kellie R Machlus[2], Marta Martínez-Bonet[1], Pui Y Lee[1,3], Alexandra Wactor[1], Nathan Nelson-Maney[1], Allyn Morris[1], Li Guo[4], Andrew Weyrich[4], Martha Sola-Visner[5], Eric Boilard[6], Joseph E Italiano[2,7], Peter A Nigrovic[1,3]*

[1]Department of Medicine, Division of Rheumatology, Immunology and Allergy, Brigham and Women's Hospital, Harvard Medical School, Boston, United States; [2]Department of Medicine, Hematology Division, Brigham and Women's Hospital and Harvard Medical School, Boston, United States; [3]Department of Medicine, Division of Immunology, Boston Children's Hospital, Harvard Medical School, Boston, United States; [4]Program in Molecular Medicine and Department of Internal Medicine, University of Utah, Salt Lake City, United States; [5]Department of Neonatology, Boston Children's Hospital, Harvard Medical School, Boston, United States; [6]Centre de Recherche en Rhumatologie et Immunologie, Centre de Recherche du Centre Hospitalier Universitaire de Québec, Faculté de Médecine de l'Université Laval, Québec, Canada; [7]Vascular Biology Program, Department of Surgery, Boston Children's Hospital, Harvard Medical School, Boston, United States

*For correspondence: pnigrovic@bwh.harvard.edu

**Abstract** Bone marrow megakaryocytes engulf neutrophils in a phenomenon termed emperipolesis. We show here that emperipolesis is a dynamic process mediated actively by both lineages, in part through the β2-integrin/ICAM-1/ezrin pathway. Tethered neutrophils enter in membrane-bound vesicles before penetrating into the megakaryocyte cytoplasm. Intracytoplasmic neutrophils develop membrane contiguity with the demarcation membrane system, thereby transferring membrane to the megakaryocyte and to daughter platelets. This phenomenon occurs in otherwise unmanipulated murine marrow in vivo, resulting in circulating platelets that bear membrane from non-megakaryocytic hematopoietic donors. Transit through megakaryocytes can be completed as rapidly as minutes, after which neutrophils egress intact. Emperipolesis is amplified in models of murine inflammation associated with platelet overproduction, contributing to platelet production in vitro and in vivo. These findings identify emperipolesis as a new cell-in-cell interaction that enables neutrophils and potentially other cells passing through the megakaryocyte cytoplasm to modulate the production and membrane content of platelets.
DOI: https://doi.org/10.7554/eLife.44031.001

## Introduction

Megakaryocytes (MKs) are the cellular source of platelets. Derived from hematopoietic stem cells, developing MKs undergo multiple rounds of endomitosis to become highly-polyploid cells averaging 20 to 100 µm in size (*Levine et al., 1982*; *Machlus and Italiano, 2013*). Mature MKs develop a complex network of intracytoplasmic membrane, termed the demarcation membrane system (DMS), that provides a membrane reservoir to enable platelet generation (*Schulze et al., 2006*). MKs then protrude pseudopodial extensions of this membrane via the marrow sinusoids into the bloodstream,

where shear stress releases fragments that become the mature platelets required for hemostasis (*Junt et al., 2007*).

Representing less than 0.3% of hematopoietic cells in bone marrow (*Levine et al., 1982*; *Machlus and Italiano, 2013*; *Winter et al., 2010*), MKs interact with other hematopoietic lineages. MKs provide a niche for plasma cells (*Winter et al., 2010*), promote neutrophil egress via production of CXCR2 ligand (*Köhler et al., 2011*), and regulate hematopoietic stem cell homeostasis (*Bruns et al., 2014*; *Zhao et al., 2014*). Almost 50 years ago, it was observed that MKs can engulf other hematopoietic cells, most commonly neutrophils (*Larsen, 1970*). Examination of fresh aspirates revealed movement of these cells within MKs, giving rise to the name emperipolesis from the Greek, *em* inside, *peri* around, *polemai* wander about (*Humble et al., 1956*; *Larsen, 1970*). Emperipolesis is observed in healthy marrow and increases with hematopoietic stress, including in myelodysplastic and myeloproliferative disorders (*Cashell and Buss, 1992*; *Mangi and Mufti, 1992*), myelofibrosis (*Centurione et al., 2004*; *Schmitt et al., 2002*; *Spangrude et al., 2016*), gray platelet syndrome (*Di Buduo et al., 2016*; *Larocca et al., 2015*; *Monteferrario et al., 2014*), essential thrombocythemia (*Cashell and Buss, 1992*), and blood loss or hemorrhagic shock (*Dziecioł et al., 1995*; *Sahebekhitiari and Tavassoli, 1976*; *Tavassoli, 1986*). Its mechanism and significance remain unknown. It has been speculated that MKs could represent a sanctuary for neutrophils in an unfavorable marrow environment, or a route for neutrophils to exit the bone marrow, but more typically emperipolesis is regarded as a curiosity without physiological significance (*Lee, 1989*; *Sahebekhitiari and Tavassoli, 1976*; *Tavassoli, 1986*).

Recently, we identified evidence for a direct role for MKs in systemic inflammation, highlighting the potential importance of the interaction of MKs with immune lineages (*Cunin and Nigrovic, 2019*; *Cunin et al., 2017*). Whereas the preservation of emperipolesis in monkeys (*Stahl et al., 1991*), mice (*Centurione et al., 2004*), rats (*Tanaka et al., 1996*), and cats and dogs (*Scott and Friedrichs, 2009*) implies evolutionary conservation, we sought to model this process in vitro and in vivo to begin to understand its biology and function. We show here that emperipolesis is a tightly-regulated process mediated actively by both MKs and neutrophils via pathways reminiscent of leukocyte transendothelial migration. Neutrophils enter MKs within membrane-bound vesicles but then penetrate into the cell cytoplasm, where they develop membrane continuity with the demarcation membrane system (DMS) to transfer membrane to MKs and thereby to platelets, accelerating platelet production. Neutrophils then emerge intact, carrying MK components with them. Together, these data identify emperipolesis as a previously unrecognized type of cell-in-cell interaction that mediates a novel form of material transfer between immune and hematopoietic lineages.

## Results

### In vitro modeling of emperipolesis reveals a rapid multi-stage process

Whole-mount 3-dimensional (3D) immunofluorescence imaging of healthy C57Bl/6 murine marrow revealed that ~6% of MKs contain at least one neutrophil, and occasionally other bone marrow cells (*Figure 1A* and *Video 1*). Emperipolesis was similarly evident upon confocal imaging of unmanipulated human marrow (*Figure 1B*). To model this process, we incubated cultured murine or human MKs with fresh bone marrow cells or peripheral blood neutrophils, respectively (*Figure 1C and D*). Murine MKs, derived either from bone marrow or fetal liver cells, were efficient at emperipolesis (~20–40% of MKs). Neutrophils were by far the most common participants, although B220+ B cells, CD115+ monocytes, and occasional CD3+ T cells and NK1.1+ NK cells were also observed within MKs (*Figure 1—figure supplement 1A*). Emperipolesis was less efficient in human cultured MKs (2–5% of MKs), which are typically smaller than murine MKs, and was observed in MKs cultured from marrow CD34+ cells but not from the even smaller MKs derived from cord blood CD34+ cells (*Figure 1D* and not shown). We elected to continue our mechanistic studies in murine MKs, principally cultured from marrow.

### Neutrophils engaged in emperipolesis penetrate into the MK cytoplasm

Confocal microscopy revealed four distinct steps. First, neutrophils become adherent to the MK surface (*Figure 2A* and *Video 2 and 3*), including to membrane protrusions we term *MK tethers* (*Figure 1—figure supplement 1B* and *Video 3*). Second, neutrophils enter MKs within membrane-

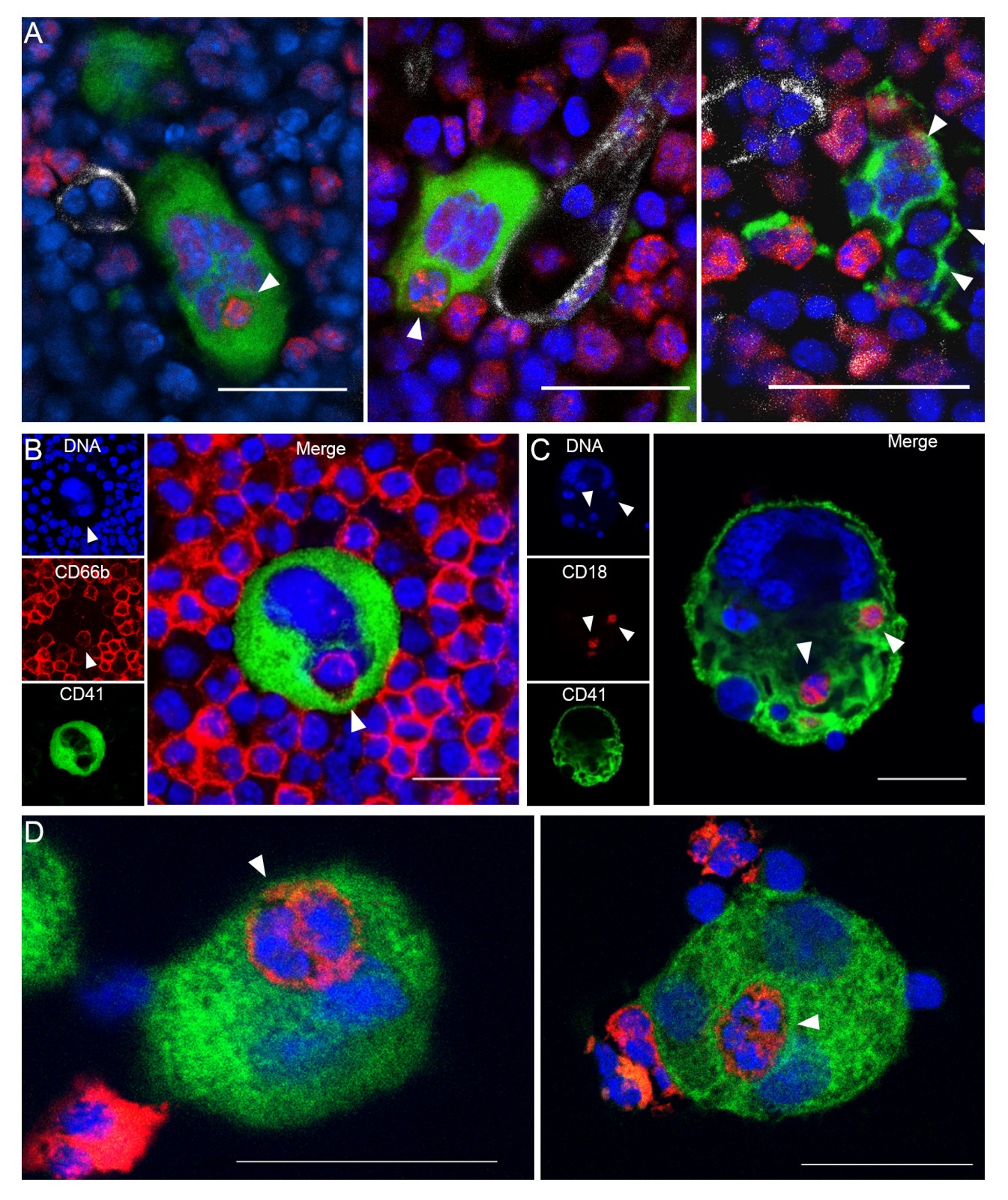

**Figure 1.** Visualization of murine and human emperipolesis by confocal microscopy. (**A**) Whole-mount images of mouse bone marrow stained with anti-CD41 (green), anti-Ly6G (red) and anti-CD31/CD144 (white). Arrowheads show internalized neutrophils or other Ly6G[neg] bone marrow cells (right image). Three-dimensional reconstitutions and confirmation of cell internalization are shown in *Video 1*. (**B**) Cells from human bone marrow aspirate were stained with anti-CD41 (green) and anti-CD66b (red). (**C**) Murine MKs were co-cultured with marrow cells overnight. Cells were stained with anti-

*Figure 1 continued on next page*

*Figure 1 continued*

CD41 (green) and anti-CD18 (red). (D) Human MKs generated from marrow CD34+ cells were co-cultured with circulating neutrophils overnight. Cells were stained with anti-CD41 (green) and anti-CD15 (red). (A-D) DNA was visualized with Draq5 or Hoechst (blue), arrowheads represent internalized neutrophils, scale bars represent 20μm, representative of at least 3 independent experiments.

DOI: https://doi.org/10.7554/eLife.44031.002

The following figure supplement is available for figure 1:

**Figure supplement 1.** Visualization of murine and human emperipolesis by confocal microscopy.

DOI: https://doi.org/10.7554/eLife.44031.003

bound vacuoles, hereafter termed *emperisomes*, bearing the MK surface marker CD41+ (*Figure 2A* and *Video 3*). Third, the emperisome undergoes transformation such that CD41 is no longer evident surrounding the neutrophil (*Figure 2A* and *Video 4*). While most MKs engaged in emperipolesis contained only one or two neutrophils, some resembled 'reservoirs' containing dozens of neutrophils in stages 2 and 3 (*Figure 1—figure supplement 1C*), an appearance recognized in human marrow as well (*Cashell and Buss, 1992*; *Larsen, 1970*; *Monteferrario et al., 2014*; *Thiele et al., 1984*) (*Figure 1—figure supplement 1D*). Fourth, neutrophils exited MKs, returning to the extracellular milieu as viable motile cells (*Figure 2A*, *Figure 1—figure supplement 1D* and *Videos 3–5*). Live cell imaging of murine MKs co-incubated with fresh bone marrow cells showed that neutrophil transit was of variable duration, in some cases lasting only a few minutes (*Figure 1—figure supplement 1E*; *Figure 4—figure supplement 1D* and *Videos 2*, *3* and *5*) and in others more than one hour (*Video 6*).

To better understand the stages of emperipolesis, we employed electron microscopy (EM). After neutrophil uptake into the emperisome (*Figure 2B*), the vacuolar space between neutrophil and MK was resorbed such that neutrophil and MK membranes became closely apposed, resulting in a structure composed of two membrane leaflets surrounding the neutrophil (*Figure 2C and D*). This structure was often associated with the appearance of neutrophilic protrusions deeper into the host MK (*Figure 2C*). Areas in which the membranes approximated very closely, becoming indistinct for short stretches, were sometimes observed (*Figure 2D*). Subsequently, only a single bilipid membrane came to separate the neutrophil cytoplasm from the MK cytoplasm, a finding that echoed the loss of CD41 staining observed by immunofluorescence, confirming dissolution of the emperisome and thereby translocation of the neutrophil to an intracytoplasmic location (*Figure 2E*). Whereas CD18 and Ly6G but not CD41 were preserved (*Figure 2A* above), this remaining membrane is most likely primarily of neutrophil origin.

## Emperipolesis is mediated by active actin cytoskeleton rearrangement in both megakaryocyte and neutrophil

To assess the cytoskeletal processes underlying this intriguing cell-in-cell interaction, we employed targeted inhibitors. The microtubule polymerization inhibitor nocodazole showed a negligible effect, but emperipolesis was dramatically curtailed by inhibitors of actin polymerization, cytochalasin D and latrunculin A (*Figure 3A*). This effect was observed when either MKs or marrow cells were exposed to these inhibitors, confirming obligate active cytoskeletal engagement by both participants (*Figure 3B*, controls of actin inhibition in *Figure 3—figure supplement 1A*). Consistent with these results, neutrophils entering MKs exhibited a polarized appearance, while MKs developed a transcellular cup similar to that observed in endothelial cells during transendothelial migration of

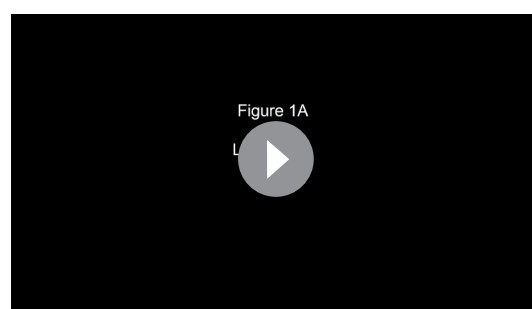

**Video 1.** Emperipolesis within murine bone marrow. Three-dimensional reconstitution of murine marrow, showing MKs (green) neutrophils (red), bone marrow sinusoids (white), and DNA (blue). Green, red or blue fluorescence are removed occasionally to visualize neutrophils inside MK or MK tethers. The three animations correspond to the three images shown in *Figure 1A*.

DOI: https://doi.org/10.7554/eLife.44031.004

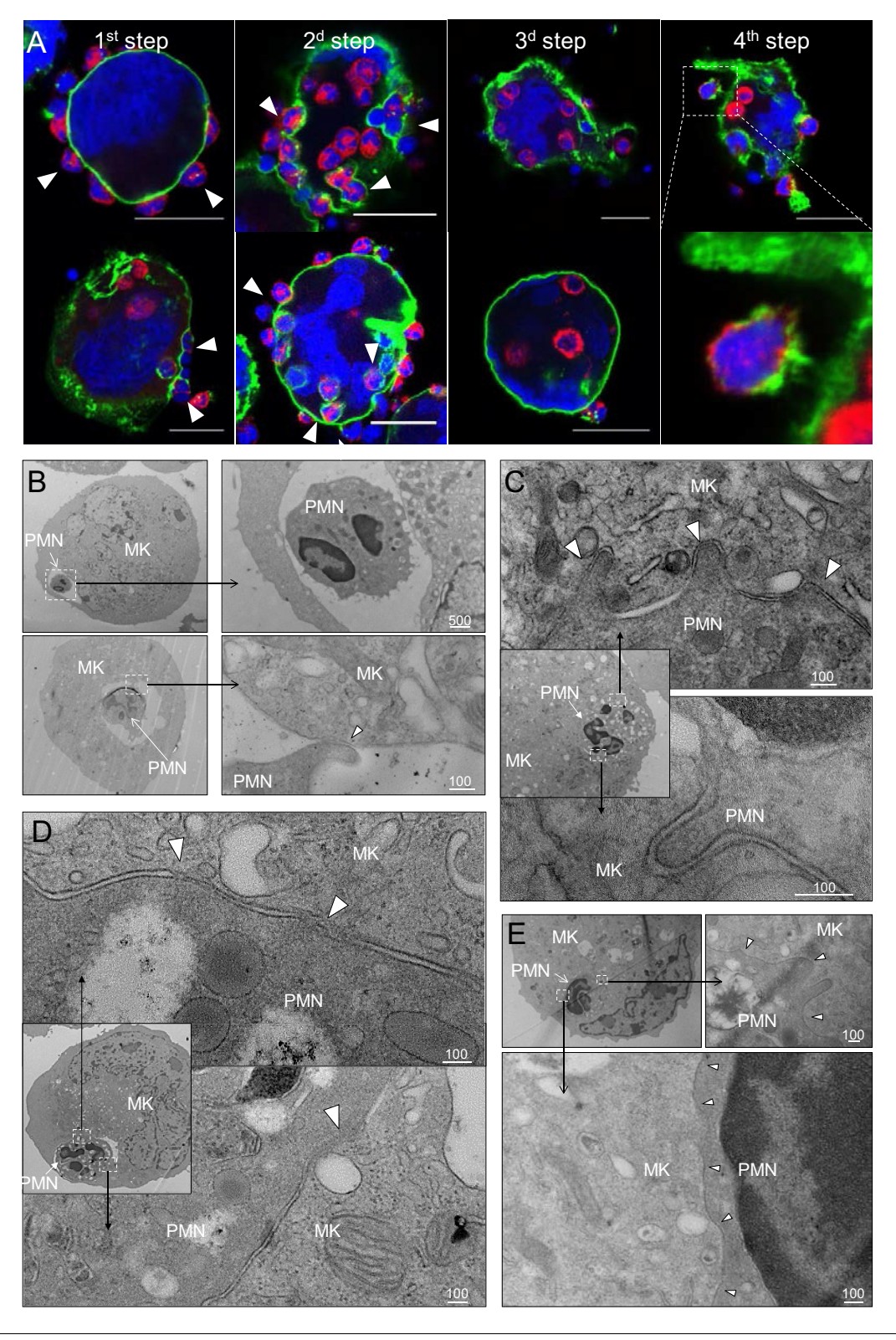

**Figure 2.** In vitro modeling of emperipolesis reveals a rapid multi-stage process. (**A**) Confocal images showing different steps of emperipolesis. Green: CD41, Red: CD18, Blue: DNA. Scale bars represent 20μm. (**B-E**) Cells were stained with OsO4 after emperipolesis assay for electron microscopy observation. (**B**) Neutrophils inside MK vacuoles. Some neutrophil surface protrusions make contact with the emperisome membrane (arrowhead). (**C-D**)
*Figure 2 continued on next page*

*Figure 2 continued*

Close interaction between neutrophil and emperisome membrane forming a two-membrane leaflet structure. (**C**) Protrusions from neutrophils that extend inside the MK cytoplasm (arrowheads). (**D**) Demarcation between emperisome membrane and neutrophil membrane disappears (arrowheads). (**E**) Neutrophil and MK are separated by a single membrane (arrowheads). (**B-E**) Scale bars in nm, representative of at least 3 independent experiments.
DOI: https://doi.org/10.7554/eLife.44031.005

leukocytes (*Carman and Springer, 2004*; *Ley et al., 2007*) (*Figure 1—figure supplement 1E*, *Figure 3—figure supplement 1B*, and *Video 2*). In agreement with inhibitor findings, actin but not microtubules localized to the interface between MKs and extracellular neutrophils and was observed to encase neutrophil-containing CD41+ vacuoles (*Figure 3D* and *Figure 3—figure supplement 1C*); by contrast, actin was not observed surrounding neutrophils that were no longer delimited by CD41-expressing membrane (*Figure 3C*). These observations demonstrate that emperipolesis is an active process mediated by actin cytoskeletal rearrangement of both MKs and neutrophils.

## Emperipolesis is mediated in part through β2-integrin/ICAM-1/ezrin

To define how MKs and neutrophils interact, we tested several candidate ligand/receptors pathways. Blocking antibodies targeting P-selectin, glycoprotein VI (GPVI), PECAM-1, CD44, and CXCR2 had no effect (not shown). However, blockade of the β2 integrin CD18, expressed by neutrophils but not MKs, strongly impaired emperipolesis (*Figure 3D*). Correspondingly, CD18-deficient bone marrow cells exhibited reduced emperipolesis into WT MKs (*Figure 3E*), despite the heightened proportion and density of neutrophils in these marrows (*Horwitz et al., 2001*).

β2 integrins bind ICAM-1, among other targets (*Ley et al., 2007*). Confocal microscopy showed that ICAM-1 is expressed by a population of human and murine MKs (*Figure 3—figure supplement 2A–D*). In agreement with previous observations in rat (*Tanaka et al., 1997*), emperipolesis by ICAM-1-deficient MKs was significantly impaired (*Figure 3F*). In further support of this mechanism, we evaluated the role of ezrin, which mediates the attachment of the intracellular tail of ICAM-1 to the actin cytoskeleton (*Heiska et al., 1998*; *Ley et al., 2007*). Inhibition of ezrin phosphorylation impaired emperipolesis (*Figure 3G*, controls of ezrin inhibition in *Figure 3—figure supplement 2E*). By confocal microscopy, ezrin could be detected only at sites of MK contact with tethered neutrophils, where it co-localized strongly with ICAM-1, consistent with its role as a bridge to the cytoskeleton (*Figure 3H and I*). By contrast, ezrin could not be visualized in MKs not tethered to leukocytes, or with only internalized leukocytes (*Figure 3H*). Together, these data show that emperipolesis is mediated in part by an interaction between neutrophil β2 integrins and MK ICAM-1/ezrin during neutrophil entry. However, absence or blockade of these factors resulted in only partial impairment of emperipolesis, indicating a role for alternate mechanisms not yet defined.

## Emperipolesis mediates membrane transfer from neutrophil to megakaryocyte

As observed by others (*Centurione et al., 2004*; *Thiele et al., 1984*), neutrophils engaged in emperipolesis frequently localized to the DMS, the intracytoplasmic membrane network implicated in platelet production (*Figure 4A*). Close examination of this interaction demonstrated membrane contiguity between the neutrophil and the DMS, suggesting that neutrophils might be able to serve as membrane donors to MKs and potentially to platelets (*Figure 4A*). To test this possibility, we employed membrane labeling. Marrow cells were stained with the lipophilic dye CellVue maroon and then co-cultured with unstained MKs. Confocal microscopy showed

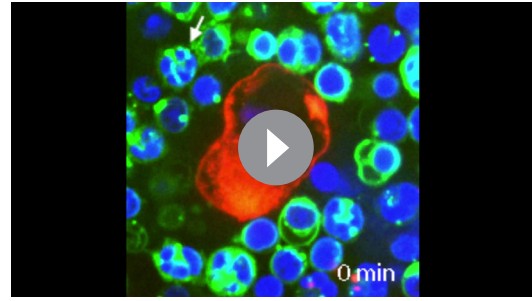

**Video 2.** Neutrophil enters megakaryocyte - formation of a trans-megakaryocyte cup. MKs stained with PKH67 (green) were co-cultured with marrow cells stained with PKH26 (red) in the presence of Draq5 (DNA, blue). Video shows the formation of a transcellular cup on the MK surface allowing neutrophil entry.
DOI: https://doi.org/10.7554/eLife.44031.006

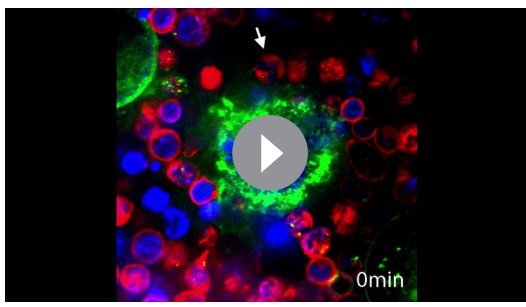

**Video 3.** Megakaryocyte tethers and neutrophil entry into a CD41+ vacuole. MKs stained with anti-CD41 (green) were co-cultured with marrow cells from mT/mG mice (red) in the presence of Draq5 (DNA, blue). Video shows a neutrophil on the MK surface, attached by MK tethers, followed by a rapid entry through a CD41+ membrane. A few minutes after its entry, the neutrophil exits at the bottom of the field of view.
DOI: https://doi.org/10.7554/eLife.44031.007

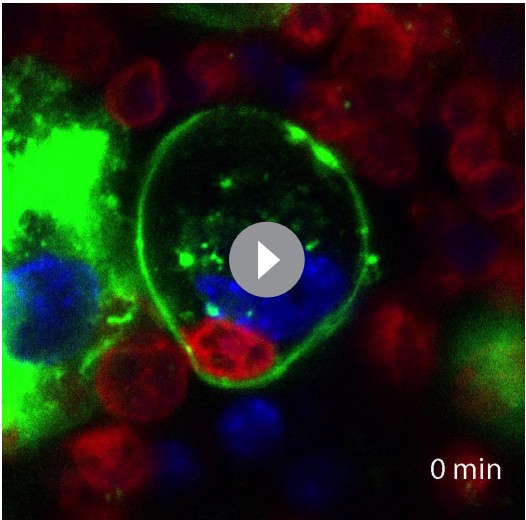

**Video 4.** Neutrophil exits megakaryocyte. MKs stained with anti-CD41 (green) were co-cultured with marrow cells from mT/mG mice (red) in the presence of Draq5 (DNA, blue). A neutrophil is present inside MK, but without interposed CD41+ membrane. Neutrophil exits MK and remains attached on its surface. Video obtained with a laser scanning confocal microscope.
DOI: https://doi.org/10.7554/eLife.44031.008

substantial loss of CellVue maroon staining in neutrophils inside but not outside MKs (*Figure 4B*). Similar loss of fluorescence was observed with the lipid stains Bodipy and PKH67, excluding a non-specific chemical interaction (*Figure 4—figure supplement 1A and B*). Further, some cells exhibited diffusion of lipid stain within the MK (*Figure 4B* and *Figure 4—figure supplement 1C*), consistent with the membrane transfer implied by EM. Finally, time-lapse spinning disk confocal microscopy confirmed transfer of neutrophil membrane to MKs from this intracellular location (*Figure 4—figure supplement 1D* and *Videos 5* and *6*). Of note, membrane transfer during emperipolesis was not associated with phosphatidylserine externalization onto the neutrophil surface (*Figure 4—figure supplement 1E–F*).

To assess reciprocal membrane transfer from MKs to neutrophils, we stained MKs with lipid stains as above and co-cultured these with unstained marrow cells. MK-derived lipids strongly co-localized with neutrophil membrane during emperipolesis (*Figure 4—figure supplement 1G–I*), while time-lapse spinning disk microscopy also confirmed reciprocal membrane exchange (*Figure 4—figure supplement 1D* and *Video 5 and 6*).

While lipid exchange from neutrophils to MKs was strongly inhibited by latrunculin A, transfer from MKs to neutrophils was not (*Figure 4C*), suggesting that this reciprocal transfer was not mediated primarily by emperipolesis. MKs produce microparticles in great abundance (*Cunin et al., 2017*; *Flaumenhaft et al., 2009*), and PKH67-stained MKs were observed to release many PKH67 +microparticles in a latrunculin A-independent manner that could transfer membrane fluorescence to neutrophils in the absence of intact MKs (*Figure 4—figure supplement 1J*). By contrast, no fluorescence was detected on MKs cultured with supernatant from PKH67-stained marrow cells, rendering unlikely a role of marrow cell-derived microparticles, exosomes or apoptotic bodies in membrane transfer

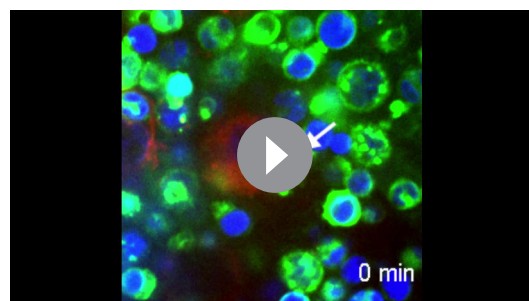

**Video 5.** Transfer of membrane during rapid emperipolesis. MKs stained with PKH67 (green) were co-cultured with marrow cells stained with PKH26 (red) in the presence of Draq5 (DNA, blue). Video shows a neutrophil entering and rapidly transiting through a MK, leaving green membrane behind. Green or red fluorescence is removed at some time points to visualize bi-directional membrane transfer.
DOI: https://doi.org/10.7554/eLife.44031.009

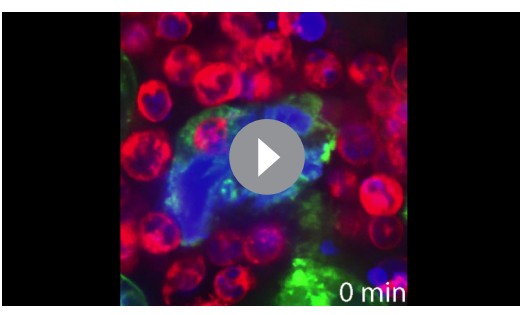

**Video 6.** Transfer of membrane during long-lasting emperipolesis. MKs stained with PKH67 (green) were co-cultured with marrow cells stained with PKH26 (red) in the presence of Draq5 (DNA, blue). Video shows a neutrophil residing within an MK. Green or red fluorescence is removed at some time points to visualize bi-directional membrane transfer.
DOI: https://doi.org/10.7554/eLife.44031.010

from neutrophils to MKs (*Figure 4—figure supplement 1K*). Thus, neutrophils transfer plasma membrane to MKs via emperipolesis, while reciprocal membrane exchange from MKs to neutrophils likely occurs primarily via MK microparticles, a phenomenon of interest not explored further here.

We then sought to determine whether membrane transfer mediates exchange of surface proteins. We performed surface biotinylation of MKs and marrow cells, and then co-cultured these cells with unstained marrow cells or MKs, respectively. Using streptavidin, we could not detect biotin on neutrophils incubated with biotinylated MKs, suggesting the absence of bulk surface protein transfer from MKs to neutrophils (not shown). However, surface biotin could be detected on some MKs after incubation with biotinylated marrow cells (*Figure 4D*), confirming that membrane exchange from neutrophils to MK transfers proteins. The nature of these proteins remains to be determined since MKs remained negative for hallmark neutrophil proteins such as CD18 and Ly6G (not shown).

## Neutrophil membranes transferred in emperipolesis emerge on circulating platelets

Platelets are generated by MKs via the DMS network, an impressively extended network of membrane whose biogenesis remains incompletely understood (*Eckly et al., 2014*; *Schulze et al., 2006*). EM had demonstrated membrane continuity between cytoplasmic neutrophils and the DMS. We therefore tested whether emperipolesis could transfer neutrophil membrane to platelets. MKs require shear stress for physiological platelet biogenesis, rendering the in vivo context most suitable for these studies. MKs stained with the cytoplasmic dye Green-CMFDA were incubated with marrow cells stained with the lipid marker CellVue Maroon and then engrafted intravenously into congenic recipient mice, in which production of CMFDA+ platelets was monitored by serial phlebotomy (*Cunin et al., 2017*; *Fuentes et al., 2010*; *Zhang et al., 2016*) and *Figure 5—figure supplement 1A*). Remarkably, most platelets produced by donor MKs (i.e. CMFDA+) were also positive for Cell-Vue Maroon, indicating a high frequency of incorporation of donor leukocyte membrane (*Figure 5A*). The intensity of CellVue Maroon staining remained constant over time, suggesting that donor membrane was employed continuously over an extended period (*Figure 5B*). Similar findings were obtained with lipid stainer PKH67 (*Figure 5—figure supplement 1B*). To exclude experimental artifact related to lipid stains, we employed donor marrow from mT/mG mice bearing membrane fluorescence mediated by fluorochrome associated with the inner membrane leaflet. Confocal imaging confirmed that membrane fluorescence from mT/mG marrow cells efficiently transferred into MKs in vitro (*Figure 5—figure supplement 1C*). Membrane fluorescence was also detected on platelets produced in vivo by WT MKs incubated with mT/mG marrow donors, albeit with weaker signal since membrane fluorescence is less intense that with lipid stains (*Figure 5—figure supplement 1D*). We similarly investigated transfer of intracellular or surface protein. Marrow cells were stained with the intracellular protein stain CellTrace Violet and then co-cultured with CMFDA +MKs. Interestingly, platelets emerging in vivo contained CellTrace violet, consistent with cytoplasmic protein transfer (*Figure 5C and D*). Together, these results demonstrate that lipids and intracellular proteins are transferred from marrow cells not only to MKs but also to their daughter platelets. Of note, we could not detect biotin on emerging platelet when MKs were previously co-cultured with surface-biotinylated marrow cells (*Figure 5—figure supplement 1E*). Moreover, platelets were negative for neutrophil surface proteins Ly6G, CD11b, CD18 and CD88 (not shown). We cannot exclude the possibility that other surface proteins not directly assessed may still transfer in quantities too modest to be detected by bulk biotin-streptavidin staining.

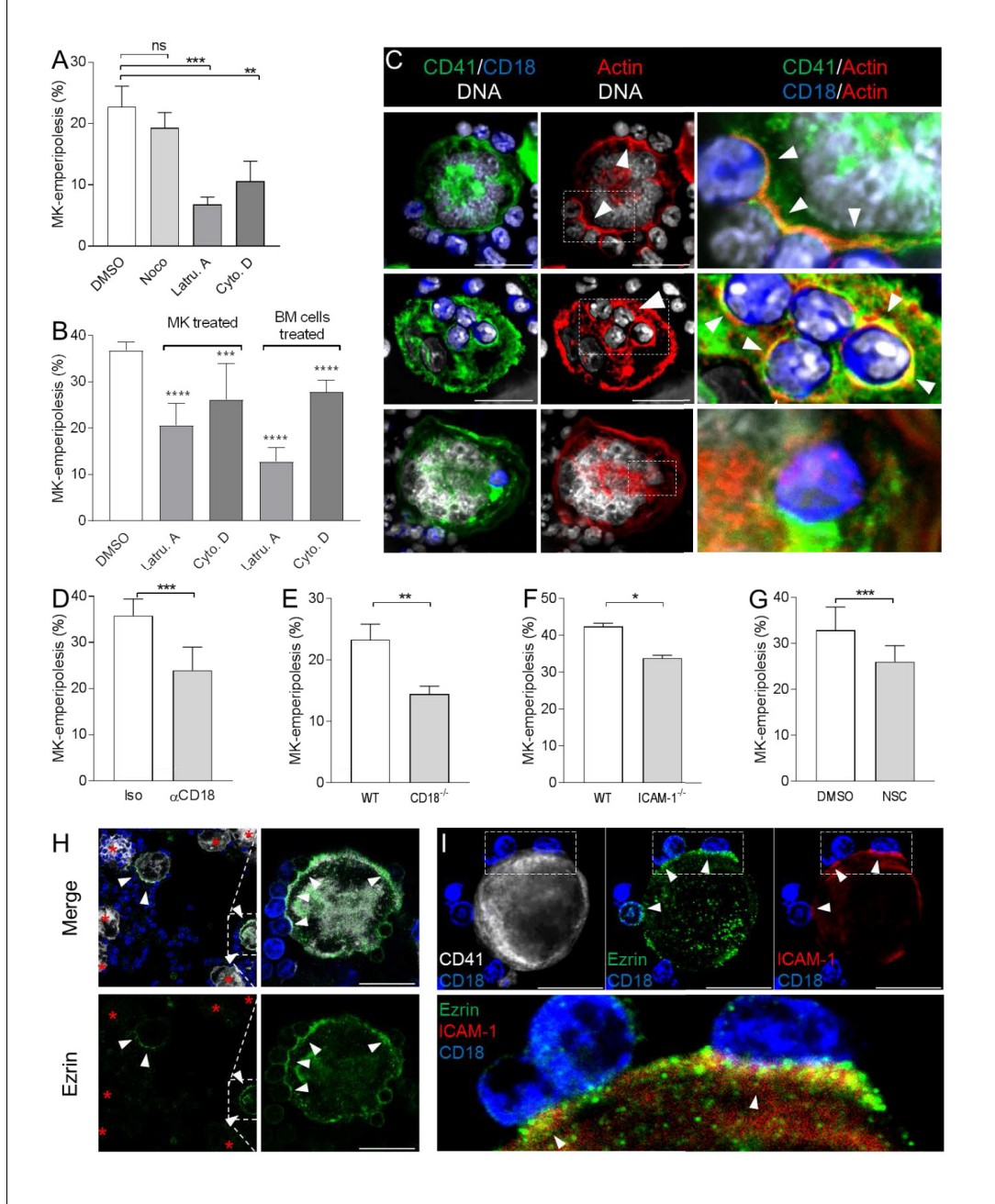

**Figure 3.** Emperipolesis is mediated by active actin cytoskeleton rearrangement and in part through β2-integrin/ICAM-1/ezrin. (A) MKs and marrow cells were co-cultured in the presence of 1μM nocodazole, latrunculin A, cytochalasin D, or control vehicle. (B) MKs or marrow cells were treated with latrunculin A or cytochalasin D for two hours. After thorough washings, cells were co-cultured with untreated marrow cells or MKs, respectively. (A-B) Cells are stained with anti-CD41, -CD18 or -Ly6G and Draq5, and observed by confocal microscopy. Histograms show percentages of MKs containing at least one neutrophil. At least 150 (A) or 500 (B) MKs per condition were counted; pool of 3 independent experiments (See *Figure 3—source data 1*) (C) Cells are stained with anti-CD41 (green), anti-CD18 (blue) and phalloidin (red). DNA is visualized with Hoechst (gray). Images show F-actin on MK surface where neutrophils are attached (upper photos), around neutrophils encapsulated in CD41+ vacuoles (middle photos) or free within MKs (lower photos). (D-G) Emperipolesis assay was performed (A) in the presence of 10μg/ml anti-CD18 or corresponding isotype control rat IgG1 (B) using marrow cells from WT versus CD18-deficient mice or (C) using MKs from WT versus ICAM-1-deficient mice, or (D) in the presence of 1μM of ezrin inhibitor NSC668394. (A-D) Histograms show percentages of MKs containing at least one neutrophil. At least 350 MKs per condition were counted; pool of 2 (F), 3 (E), or 4 (D and G) independent experiments. (See *Figure 3—source data 1*) H. After co-culture, cells are stained with anti-CD41 (white), anti-ezrin (green) and anti-Ly6G (blue). Arrows show ezrin clustering on the MK surface. Red asterisks show MKs without detectable ezrin. (I) Cells are stained with anti-CD41 (gray), -ezrin (green), -ICAM-1 (red), -Ly6G (blue). Arrows show ICAM-1/ezrin co-localization on the MK surface. Lower photo is a magnification of the dashed white region. (H and I). Scale bars represent 20μm, representative of at least 3 independent experiments.

*Figure 3 continued on next page*

eLIFE Research article

Cell Biology

*Figure 3 continued*

DOI: https://doi.org/10.7554/eLife.44031.011

The following source data and figure supplements are available for figure 3:

**Source data 1.** Source data for *Figure 3*.
DOI: https://doi.org/10.7554/eLife.44031.014
**Figure supplement 1.** Emperipolesis is mediated by active actin cytoskeleton rearrangement and in part through β2-integrin/ICAM-1/ezrin.
DOI: https://doi.org/10.7554/eLife.44031.012
**Figure supplement 2.** Emperipolesis is mediated by active actin cytoskeleton rearrangement and in part through β2-integrin/ICAM-1/ezrin.
DOI: https://doi.org/10.7554/eLife.44031.013

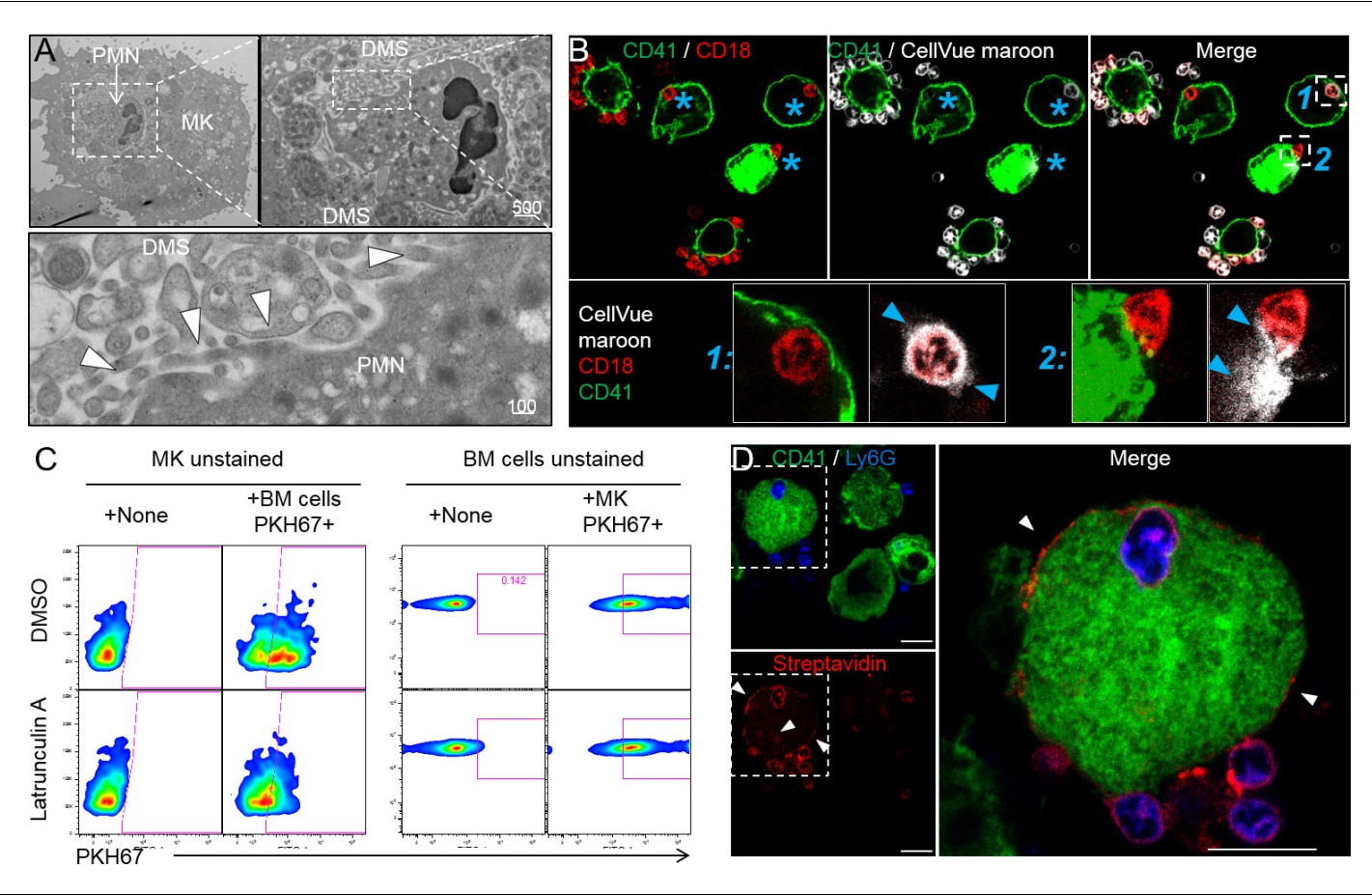

**Figure 4.** Emperipolesis mediates membrane transfer from neutrophil to megakaryocyte.  (A) Cells were stained with OsO4 after emperipolesis assay for electron microscopy observation. Images show a membrane continuity between neutrophil and DMS (arrowheads). (B) Bone marrow cells were stained with CellVue Maroon prior to co-culture with MKs. Confocal microscopy shows a loss of CellVue Maroon fluorescence in neutrophils engaged in emperipolesis (blue asterisks). Lower photos show details for the two neutrophils (1) and (2) on the upper right photo (transfer of CellVue Maroon inside MKs, blue arrowheads). (C) MKs and marrow cells were co-cultured with 1µM latrunculin A or DMSO. Left panels: marrow cells are previously stained with PKH67, dot plots show PKH67 fluorescence on CD41+ MKs. Right panels: MKs are previously stained with PKH67, dot plots show PKH67 fluorescence on Ly6G+ neutrophils. (D) Surface proteins of marrow cells were biotinylated prior to emperipolesis assay. After fixation, cells were incubated with AF594-streptavidin (red). Asterisks show the presence of biotinylated proteins on MK surface and DMS. B and D: Scale bars represents 20µm, representative of at least 3 independent experiments.

DOI: https://doi.org/10.7554/eLife.44031.015

The following figure supplement is available for figure 4:

**Figure supplement 1.** Emperipolesis mediates membrane transfer from neutrophil to megakaryocyte.
DOI: https://doi.org/10.7554/eLife.44031.016

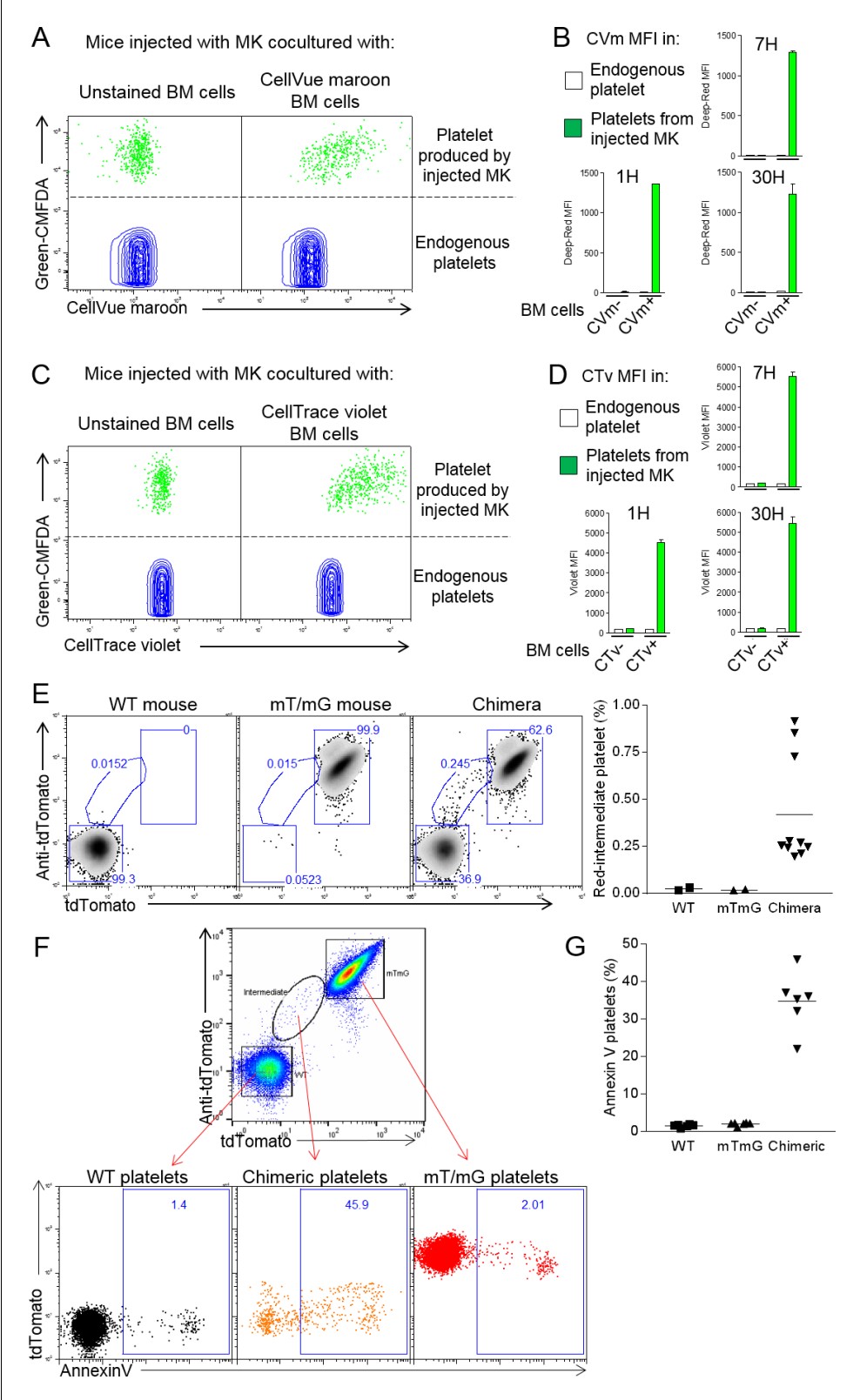

**Figure 5.** Neutrophil membranes transferred in emperipolesis emerge on circulating platelets. (**A**) CMFDA+ MKs are co-cultured with either CellVue maroon (CVm) or unstained marrow cells, and injected i.v. (***Figure 5—figure supplement 1A***). Circulating CD41+ platelets were analyzed serially by flow cytometry. Dot plots show CVm MFI among CMFDA- platelets (endogenous) and CMFDA+ platelets (produced by injected MKs). (**B**) CellVue maroon MFI on endogenous platelets versus platelets produced by injected MKs over time. (**C-D**) Same as (**B-C**) but marrow cells were stained with the

*Figure 5 continued on next page*

*Figure 5 continued*

protein stain CellTrace violet (CTv) prior to co-culture with CMFDA+ MKs. (**E-F**) CD45.1 mice were lethally irradiated and engrafted for 4 weeks with a mix of WT and mT/mG marrow. Circulating platelets were fixed, permeabilized and stained with anti-tdTomato. (**E**) Left: representative dot plots of tdTomato and anti-tdTomato MFI in circulating platelets from WT, mT/mG, and chimera mice. Right: frequency of the red-intermediate population among total platelets, representative of 3 independent experiments. Note that this population underestimates the true post-emperipolesis population because it detects only mT/mG→WT events, not WT→WT, mT/mG→mT/mG, and WT→mT/mG. (**F**) Annexin V MFI in WT, mT/mG, and chimeric platelets (control for binding specificity in *Figure 5—figure supplement 1G*). (**G**) Frequency of Annexin V positive events among WT, mTmG and chimeric platelets. (**F-G**) Representative of 3 experiments.

DOI: https://doi.org/10.7554/eLife.44031.017

The following figure supplement is available for figure 5:

**Figure supplement 1.** Neutrophil membranes transferred in emperipolesis emerge on circulating platelets.

DOI: https://doi.org/10.7554/eLife.44031.018

We sought to exclude the possibility that this membrane transfer reflected an artifact of ex vivo MK generation and co-culture. To this end, we generated mice chimeric for WT and mT/mG marrow, allowing us to seek platelets resulting from mT/mG→WT membrane transfer in a fully native environment. Indeed, platelets with the expected intermediate fluorescent phenotype were observed (*Figure 5E*), albeit only in relatively small numbers, potentially because of the weak fluorescence in the mT/mG system and because WT→WT, mT/mG→mT/mG, and WT→mT/mG transfer events remain undetectable. Examination of BM MKs identified examples of fluorescent neutrophils contributing membrane to non-fluorescent MKs from an intracellular location (*Figure 5—figure supplement 1F*). We conclude that intracellular neutrophils transfer membrane to MKs and thereby to platelets via emperipolesis in vivo. Of note, we observed an important fraction of the 'red-intermediate' platelets in the mT/mG/WT chimeric mice expressing phosphatidylserine (*Figure 5F and G*; control for Annexin V staining in *Figure 5—figure supplement 1G*), while we observe normal levels of CD62P and a marginal band of β1-tubulin by microscopy (*Figure 5—figure supplement 1H–I*), excluding an abnormal activation phenotype (*Moskalensky et al., 2018*; *Sadoul, 2015*). Surface phosphatidylserine creates a scaffold for clotting factors and is a hallmark of pro-coagulant platelets (*Heemskerk et al., 2013*; *Nagata et al., 2016*). This result suggests the possibility that emperipolesis-derived platelets could be functionally distinct, potentially including enhanced thrombogenic capacity.

## Emperipolesis enhances platelet production

As a bidirectional interaction between MKs and leukocytes, emperipolesis is likely to have multiple cellular effects. Among these, we elected to explore its impact on thrombocytogenesis. Recognizing the association of emperipolesis in humans with hematopoietic disease (*Cashell and Buss, 1992*; *Centurione et al., 2004*; *Larsen, 1970*; *Mangi and Mufti, 1992*; *Stahl et al., 1991*; *Thiele et al., 1984*), we exposed mice to several models of stress-induced platelet over-production by MKs, intraperitoneal LPS injection and IgG-mediated thrombocytopenia. The proportion of MKs containing at least one neutrophil was assessed in two-dimensional marrow sections. In each case, emperipolesis increased from a baseline of ~2–5% in control mice to ~6–10% under stress (*Figure 6A–C*). These figures represent a minimal estimate of the 'snapshot' prevalence of emperipolesis, since they sample only 5 μm sections of MKs with a typical diameter of 20–100 μm, but nevertheless confirm that emperipolesis is common and strongly induced under physiological stress. Interestingly, an enhanced drive for platelet production was not sufficient to augment emperipolesis, because accelerated platelet production following administration of thrombopoietin, or platelet depletion by anti-CD41 was unaccompanied by an increase in emperipolesis (*Figure 6—figure supplement 1A and B*). One possible explanation is that neutrophil activation may also be required, consistent with the role of neutrophil β2 integrins defined above.

To quantitate the impact of emperipolesis on thrombocytopoiesis, we employed IncuCyte high-content live-cell microscopy (*Thon et al., 2012*), comparing pro-platelet generation by MKs cultured alone or together with marrow cells. These studies employed fetal liver MKs because of their superior ability to generate pro-platelets in vitro. To assess the role of cell-cell contact and bone marrow cell-derived soluble factors, including microparticles, we cultured MKs with marrow cell supernatant or with paraformaldehyde-fixed marrow cells. Co-culture with living marrow cells markedly enhanced

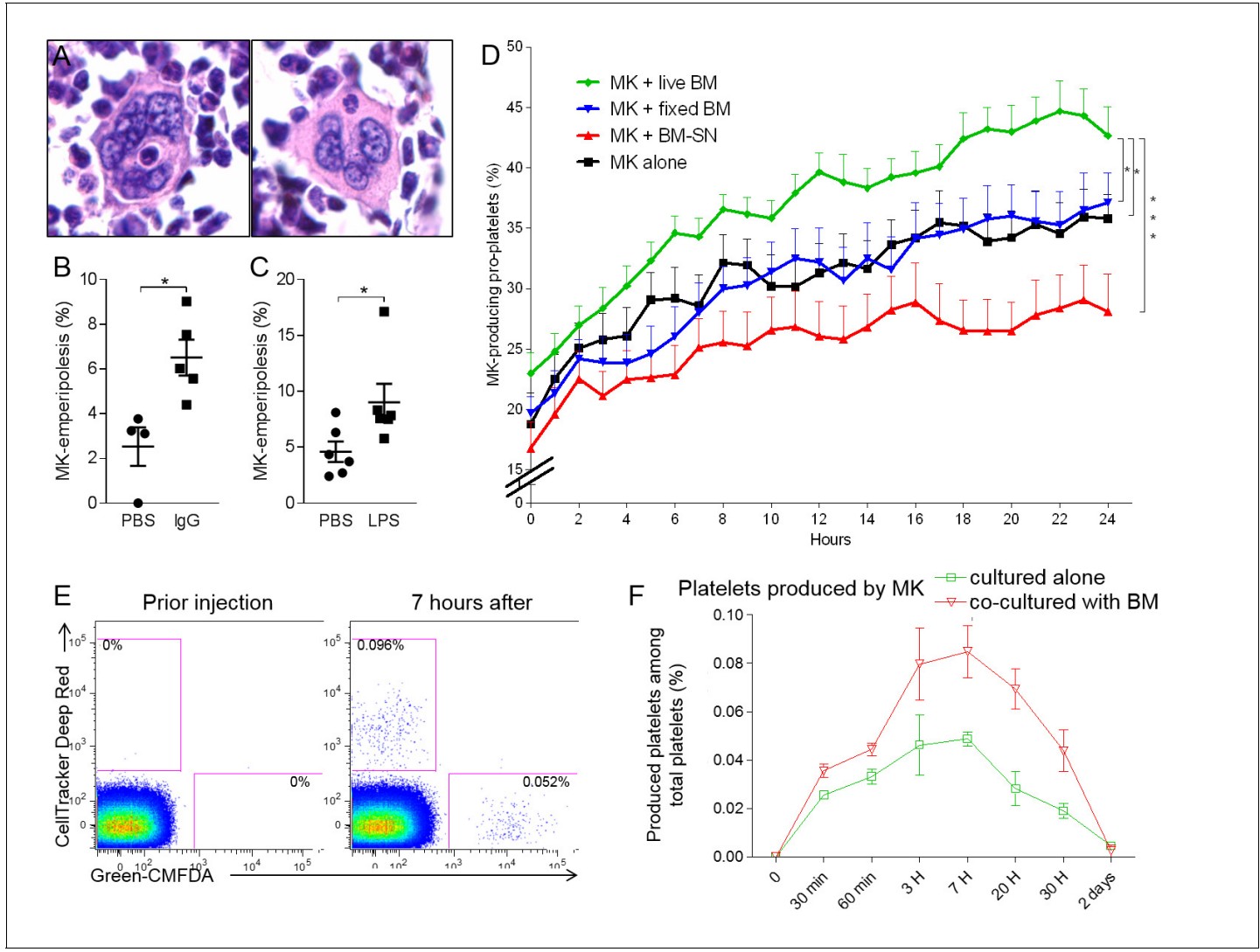

**Figure 6.** Emperipolesis contributes to platelet production.  (A) Bone sections stained with H and E showing MK containing one neutrophil in an emperiosome (left) or without evident vacuole (right). (B) Tg-FcγR2a mice are treated with HA-IgG for 7 days. Percentage of MKs containing at least one neutrophil was calculated in bone sections, n=4-5 mice per group (See *Figure 6—source data 1*). (C) Mice were treated with PBS or 25μg LPS i.p. After 3 days, the percentage of MKs containing at least one neutrophil is calculated in bone sections, n=6 mice per group (See *Figure 6—source data 1*). (D) MKs were cultured alone (black line) or with SN from marrow cell culture (red line), fixed marrow cells (blue line) or live marrow cells (green line). MKs were then enriched and cultured in TPO medium. MKs producing pro-platelets over time were quantitated using IncuCyte. None vs. live marrow: p=0.015; marrow-SN vs. live marrow: p=0.0002; fixed marrow vs. live marrow: p=0.014. Pooled from 5 independent experiments (See *Figure 6—source data 1*). (E-F) MKs stained with Green-CMFDA or CellTracker Deep Red are either co-cultured or not with marrow cells, mixed together 1:1 and injected i.v. (*Figure 6—figure supplement 1C*). After different time points, circulating platelets positive for Green-CMFDA or CellTracker Deep Red are detected by flow cytometry. (E) Representative dot plots of green vs. red staining on CD41+ platelets. (F) Frequency of green vs. red population of platelets over time, n=3 mice per group, representative of 4 independent experiments (See *Figure 6—source data 1*).

DOI: https://doi.org/10.7554/eLife.44031.019

The following source data and figure supplement are available for figure 6:

**Source data 1.** Source data for *Figure 6*.
DOI: https://doi.org/10.7554/eLife.44031.021
**Figure supplement 1.** Emperipolesis contributes to platelet production.
DOI: https://doi.org/10.7554/eLife.44031.020

pro-platelet production (*Figure 6D*). By contrast, MKs cultured with fixed cells or cell supernatants produced fewer pro-platelets than those cultured alone, weighing against a role for contact and soluble factors and implicating emperipolesis directly (*Figure 6D*).

Finally, we tested the effect of emperipolesis on platelet production in vivo via adoptive transfer. MKs were labeled either with Green-CMFDA or with CellTracker Deep Red, and one population or the other (varied across experiments) was cultured together with marrow cells. MKs were mixed 1:1 and engrafted IV into recipient animals for serial parallel quantitation of green and red platelets (*Figure 6—figure supplement 1C*). As predicted by the IncuCyte findings, MKs cultured with marrow cells were more efficient at producing platelets (*Figure 6E and F*), consistent with promotion of thrombocytogenesis by emperipolesis.

## Discussion

Megakaryocytes anchor hemostasis via elaboration of platelets. Platelet production can occur in a cell-intrinsic manner, as for example by MKs cultured in isolation ex vivo. However, physiological platelet generation proceeds in a complex multicellular environment. The present studies establish a pathway through which this cellular context modulates thrombocytogenesis. During emperipolesis, neutrophils and other hematopoietic lineages penetrate into the MK cytoplasm, a process mediated actively by both host and donor. This process is distinct from phagocytosis since the neutrophil actively penetrates into the MK and survives to exit intact. Cytoplasmic neutrophils transfer membrane and cytosolic contents to MKs and to platelets, thereby enhancing platelet production. Donor neutrophils receive membrane in turn before they exit intact (*Figure 7*). Thus, emperipolesis represents a previously unrecognized pathway through which neutrophils and other hematopoietic cells engage with MKs to modulate the composition and production of circulating platelets.

We identified β2 integrins and MK ICAM-1/ezrin as contributors to emperipolesis. These proteins also mediate another form of transcellular passage, the migration of neutrophils through the cell bodies of endothelial cells (*Ley et al., 2007*). Unlike emperipolesis, endothelial transcellular migration is not known to involve penetration into the host cytoplasm. It remains unknown whether other mechanisms are shared between emperipolesis and transendothelial migration, such as fusion of caveolin vesicles to create an intracellular channel for passage (*Ley et al., 2007*; *Millán et al., 2006*). Tavassoli and colleagues had previously postulated that emperipolesis could represent a pathway of neutrophil egress from the bone marrow (*Dziecioł et al., 1995*; *Tavassoli, 1986*). Transit of some neutrophils through MKs over the course of just a few minutes lends plausibility to this hypothesis. Our data do not exclude the possibility that some neutrophils pass through MKs without a cytoplasmic 'detour,' thereby resembling endothelial transcellular migration even more closely.

Mechanisms of MK-emperipolesis have remained entirely obscure for almost 50 years (*Larsen, 1970*). Electron microscopy observations previously raised the possibility that neutrophils are not internalized by MKs but rather enter directly through the DMS, which is continuous with the cell surface (*Eckly et al., 2014*), to reside within DMS dilated cavities (*Breton-Gorius and Reyes, 1976*; *Thiele et al., 1984*). Consistent with these observations, myeloid cells are often found at the cell surface entrance of the DMS (*Thiele et al., 1984*), and increased emperipolesis has been reported in models with dilated and enlarged DMS or after pharmacological modification of the DMS (*Overholtzer and Brugge, 2008*). However, our confocal and electron microscopy images demonstrate that neutrophils enter MKs directly, through a vacuole, ultimately taking up residence inside the MK cytoplasm. Emperipolesis is nevertheless strikingly heterogeneous. For example, emperipolesis can be observed in MKs of all sizes and can last just minutes to over an hour. MKs may enclose a single neutrophil or encompass as many as 50 neutrophils (*Figure 1—figure supplement 1C*). These observations strongly suggest that there may be different types of emperipolesis, involving different molecular pathways and serving distinct functions that remain to be defined.

Because the mechanistic pathways identified in our study are not specific to emperipolesis (e.g. β2 integrin binding, actin polymerization), we have so far been unable to interrupt this phenomenon with selectivity in vivo. To address its function, we employed a combination of approaches, focused here on platelet production. This focus is justified by the enhanced frequency of emperipolesis in diseases associated with high platelet count, including essential thrombocythemia, polycythemia vera (*Cashell and Buss, 1992*; *Vytrva et al., 2014*), or with high platelet demand (gray platelet syndrome, blood loss or hemorrhagic shock [*Di Buduo et al., 2016*; *Dziecioł et al., 1995*;

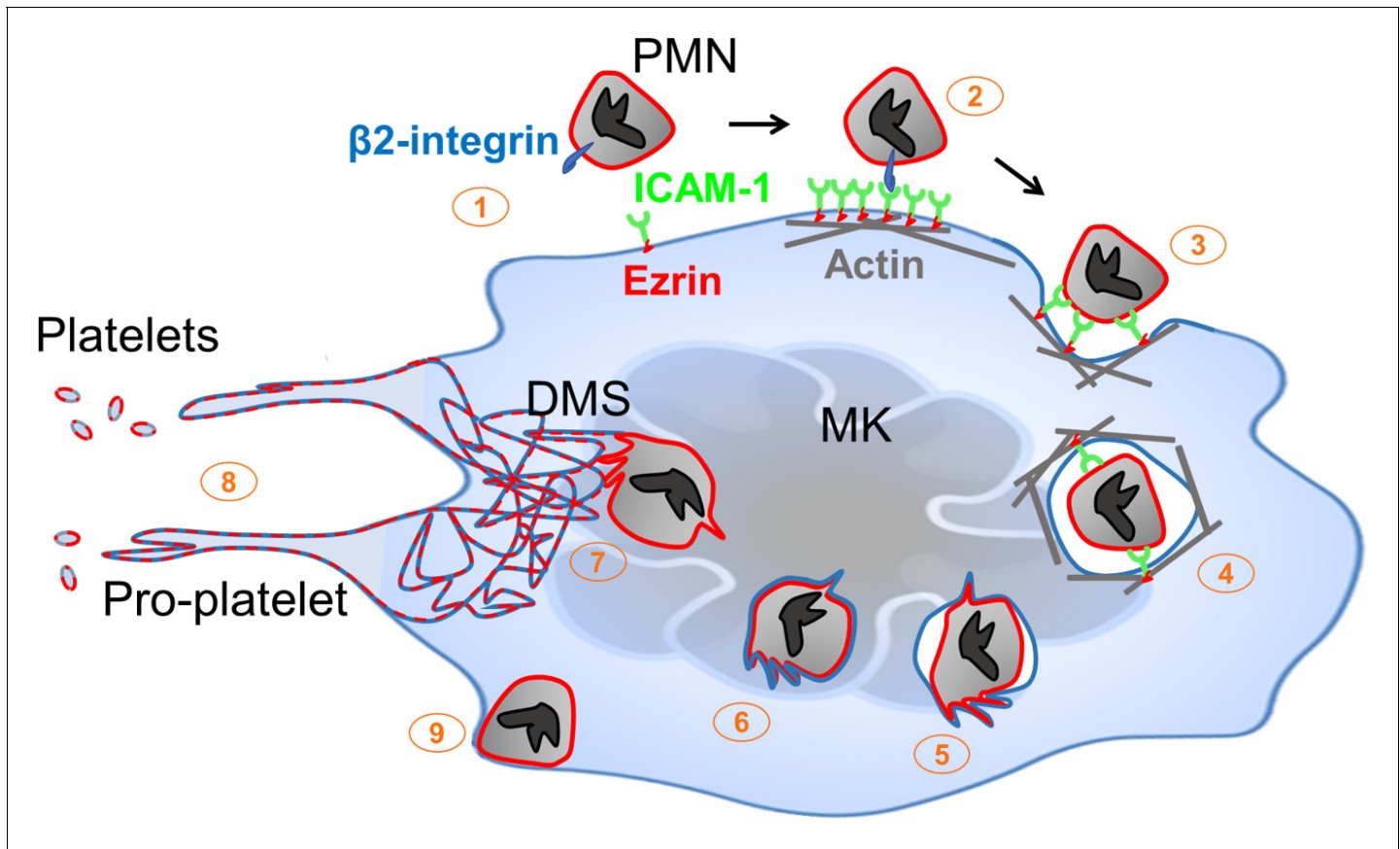

**Figure 7.** Neutrophil transit through megakaryocytes in emperipolesis mediates membrane transfer and enhanced platelet production – proposed model. (1) Neutrophils expressing β2 integrin and MKs expressing ICAM-1 are positioned to engage, including via MK tethers. (2) ICAM-1 ligation by neutrophil β2-integrins induces ICAM-1/ezrin pair translocation and clustering, as in transendothelial migration (*Ley et al., 2007*). (3) Formation of a trans-MK cup in an actin-dependent manner. (4) Neutrophil entry into MK vacuole ("emperisome"). (5-6) Neutrophil and emperisome membranes become closely apposed. Neutrophil and emperisome membranes may merge at some locations. Emperisome membrane disappears, enabling neutrophil entry into the MK cytoplasm. (7) Neutrophil translocates to DMS where membrane fusion occurs, transferring neutrophil membrane and protein. (8) Production of platelets containing neutrophil membrane. (9) Egress of viable neutrophil from MK.
DOI: https://doi.org/10.7554/eLife.44031.022

*Larocca et al., 2015*; *Monteferrario et al., 2014*; *Sahebekhitiari and Tavassoli, 1976*; *Tavassoli, 1986*]). Further, enhanced emperipolesis in chronic myeloproliferative disorders positively correlates with the peripheral platelet count (*Thiele et al., 1984*). We establish that emperipolesis accelerates platelet production both in vitro and in vivo. The quantitative importance of this contribution, and whether it reflects enhanced access to lipid membrane or some other mechanism, remains to be established.

Since platelets generated through emperipolesis bear donor membrane as well as parent MK membrane, it is likely that they will be different in function. Consistent with this possibility, we found enhanced expression of surface phosphatidylserine on emperipolesis-derived platelets in our chimeric WT-mT/mG mice (*Figure 5F and G*). Given the role of β2 integrins in emperipolesis, it is plausible to suspect that activated neutrophils will preferentially engage in emperipolesis, suggesting the possibility that 'angry neutrophils make angry platelets'. The identity of lipids and proteins transferred from neutrophils and other cells to MKs and platelets, and the resulting changes in cell function, will be important topics for future study.

The impact of emperipolesis is unlikely to be restricted to MKs and platelets. Our videomicroscopy data confirm that exiting neutrophils can carry MK membrane with them (*Figure 4—figure supplement 1D*), potentially translating into altered function. These effects are more difficult to study in vitro because of the capacity we identified for MKs to transfer membrane to nearby cells via

MK microparticles (*Figure 4—figure supplement 1H*). Our prior work had identified MK microparticles as potent pro-inflammatory vectors implicated in the delivery of IL-1 during systemic inflammatory disease (*Cunin et al., 2017*). The present findings thus extend the understanding of MK microparticles as signaling vectors. Further, by establishing transfer of membrane not only from donor cell to MK but also reciprocally, followed by release of viable cells back into the intercellular milieu, these studies identify emperipolesis as a mechanism through which MKs may be able to 'groom' neutrophils and other immune lineages.

We are unable at present to define conclusively the proportion of circulating platelets that bear neutrophil membrane. The mT/mG chimera experiments will not accurately reflect this fraction, because only one of 4 possible donor-host pairs yields detectable platelets (mT/mG neutrophil→WT MK) and because mT/mG fluorescence is weak, such that transfer events will likely often be invisible. We note that many platelets released by MKs co-cultured with membrane-labeled marrow donors expressed membrane label (*Figure 5*). Given the speed with which cells enter and exit MKs, a snapshot prevalence of 6% is compatible with the possibility that many or even most MKs, and many neutrophils, experience emperipolesis over time, perhaps repeatedly. If this is the case, and labeling experiments accurately reflect the efficiency of membrane transfer, then emperipolesis-derived membrane could be common in circulating platelets. Alternately, if transfer were inefficient, and/or only a subset of MKs engaged in emperipolesis, then emperipolesis-modulated platelets could represent simply a small (but potentially still functionally important) subset of the circulating pool.

We recognize other limitations to these studies. Unique functional contributions of emperipolesis-derived platelets remain to be established. The mechanisms through which neutrophils escape emperisomes to enter the cytoplasm, home intracellularly to the DMS, and then egress without violating MK outer membrane integrity remain to be defined. Cells deficient in β2 integrins retain the capacity to enter MKs, albeit with reduced efficiency, revealing that other ligand/receptor pairs can mediate entry. The signals driving enhanced emperipolesis in the setting of experimental stress, during hemorrhagic shock, and in aberrant marrow environments such as in hematopoietic malignancies, remain to be established. Despite these limitations, the current studies identify emperipolesis as a novel cell-in-cell interaction that mediates reciprocal transfer of membrane and other cellular components, defining thereby a new mechanism of interchange between immune and hematopoietic systems.

## Materials and methods

### Mice

C57Bl/6, CD45.1 B6 mice, *cd18*$^{-/-}$ mice, mT/mG mice and Tg-FcγRIIA mice were purchased from The Jackson Laboratory. LyzM-GFP mice transgenic for *FcγRIIA* (Tg-FcγRIIA) mice were backcrossed 10 times in the C57BL/6J background. Unless stated, all experiments employed male mice aged 8–12 weeks. All procedures were approved by the local animal care committee.

### Antibodies

Anti-CD61 (2C9.G2), -CD42d (1C2), -CD11a (M17/4), -CD11b (M1/70), CD62P (RMP-1), CD31 (MEC13.3, all from Biolegend), -Ly6G (1A8, BioXCell), -GPVI (Jaq1, Emfret), and anti-CD18 (clone GAME-46, BD biosciences) were used in blocking experiments. Antibodies used for flow cytometry and microscopy staining were anti mouse-CD41 (MWReg30), -ICAM (YN1/1.7.4), -CD18 (M18/2), -Ly6G (1A8), -CD31 (MEC13.3), CD144 (BV13), -Tubulin (10D8 all from Biolegend), -Ezrin and -phospho-Ezrin (rabbit polyclonal, Cell Signaling), and anti-human-CD41 (HIP8), -CD15 (W6D3), CD66b (G10F5), and -ICAM-1 (HA58). Fluorescent-conjugated secondary antibodies were purchased from Jackson Immunoresearch.

### Chemicals and reagents

Latrunculin A and Cytochalasin D were purchased from Cayman Chemical. Ezrin inhibitor NSC668394 was from Calbiochem. Lipids cells strainers PKH67 and PKH26 were from Sigma. Protein strainers Green CMFDA, CellTracker Deep Red and Cell Trace Violet were from Molecular probes. Surface protein biotinylation kit was purchased from Pierce.

## Cell generation

Marrow cell generation: marrow was flushed from the bone marrow cavity using PBS and cells were filtered via a 40 µm cell strainer to remove spicules and clumps. After centrifugation, erythrocytes were lysed and then cells were resuspended in complete RPMI medium supplemented with 1% supernatant from the TPO-producing fibroblast cell line GP122 (hereafter called TPO medium) (*Villeval et al., 1997*). Murine MK generation: Bone marrow cells were cultured in TPO medium for 4–5 days. MKs were separated from marrow cells using a two-step albumin gradient as described (*Schulze, 2016*; *Shivdasani and Schulze, 2005*). Fetal liver-derived MKs were generated as described (*Machlus et al., 2017*). Human MK generation: mobilized peripheral blood or BM CD34 +stem/progenitor cells were purchased from AllCells. $1 \times 10^5$ cells were cultured in StemSpan medium supplemented with a MK expansion supplement, both from StemCell for 12–14 days, as described (*Liu et al., 2011*). Human neutrophils were obtained from blood from healthy donors. Neutrophil were enriched using a dextran density gradient sedimentation as described (*Cunin et al., 2016*). Contaminating red blood cells were lysed by hypo-osmotic shock.

## Emperipolesis assay

$2 \times 10^4$ murine MKs with $2 \times 10^6$ murine marrow cells in TPO medium, or $2 \times \times 10^4$ human MKs with $2 \times 10^6$ human neutrophil in StemSpan medium were co-cultured overnight in P96 round bottom wells.

## Confocal microscopy

Cells were fixed in PFA 2% for 30 min at RT. After washing, cells were suspended in PBS supplemented with 0.1% saponin and 3% FCS (permeabilization buffer) and incubated 2 hr at RT or overnight at 4C with 10 µg/ml primary antibodies. After washing in permeabilization buffer, secondary antibodies diluted 1:200 were added for 1 hr. When indicated, Phalloidin (Molecular Probe), Draq5 (eBioscience) or Hoechst (ThermoFisher) were added for the 15 last minutes, prior washing with PBS and cytospin. Cells were mounted on slides using FluorSave mounting medium (Calbiochem). Microscopy was performed using a Nikon C1 Plus Confocal Laser Scanner confocal or a Zeiss LSM 710 or 800 Multiphoton Laser scanner confocal microscope.

## Spinning disk confocal microscopy

MKs stained with PKH26 and marrow cells stained with PKH67 were co-cultured in P96 round bottom wells for at least 1 hr prior to spinning disk imaging. Cells were then resuspended in TPO medium without red phenol and supplemented with Draq5, and cultured in a micro insert 4-well dish for time lapse imaging. Movies were obtained using a YokogawaCSU-X1 or an Olympus DSU inverted spinning disk confocal microscope. Images were acquired every 4 min on 12 different Z-stacks, 1 µm per stack. Movies were analyzed using EZ element software or Volocity software.

## Imaging of whole-mount bone marrow

Whole-mount-tissue preparation, immunofluorescence staining and imaging of the bone marrows were performed as described previously (*Kunisaki et al., 2013*). Briefly, mice were intravenously injected with AF647-labelled anti-CD31 and anti-CD144 and perfused with PBS and 4% PFA15 min after. Femurs and tibias were harvested, PFA-fixed, frozen in OCT, and shaved on a cryostat to expose the marrow. Bones were incubated in PBS containing 10% FCS and 0.5% Triton X-100, with AF594 anti-Ly6G, AF488 anti-CD41 and Hoechst for 2 days. Images were acquired using a Zeiss LSM 800 Multiphoton Laser scanner confocal microscope and reconstructed in 3D with Imaris software.

## Electron microscopy

Cells were fixed using 2% PFA and 0.1% glutaraldehyde for 1 hr RT. After washings, cells were incubated with 0.1% OsO4 for 30 min prior to sectioning. 50 nm sections were observed with a JEOL 1200EX electron microscope.

## In vitro pro-platelet production

MKs were co-cultured overnight without or with PKH67-stained marrow cells. Marrow cells were separated from MKs using BSA-gradient sedimentation, and cells were transferred to a P24 well plate

and imaged using the IncuCyte HD system (Essen BioScience). Frames were captured every hour. Rates and extent of proplatelet production were measured in ImageJ software using investigator-coded software (*Thon et al., 2012*).

### In vivo platelet production

$2 \times 10^5$ MKs, previously stained with Green CMFDA or CellTracker Deep Red (Molecular probes) and co-cultured or not with marrow cells, were injected i.v. in 200 µl PBS. Blood was harvested by tail vein sampling at indicated time points using heparinized capillary tubes. 1 µl was blood is diluted in 500 µl PBS in the presence of an anti-CD41 antibody. Presence of green CMFDA or CellTracker Deep Red on circulating CD41+ platelets was evaluated by flow cytometry.

### Emperipolesis in marrow sections

Bones were fixed in PFA 4% for 2 days prior to decalcification in Kristensen solution for 2 days prior to paraffin-embedding. Percentages of marrow MKs containing at least one neutrophil were determined on 6 µm paraffin-embedded sections stained with H and E.

### Platelet overproduction models

LPS treatment: WT C57Bl/6J mice were treated i.p. with PBS or 25 µg LPS in 200 µl PBS. Bones were harvested 3 days later. *FcγRIIA transgenic mice:* Tg-FcγRIIA mice were treated i.v. with PBS or 500 µg heat-agglutinated IgG. Bones were harvested 7 days later. *TPO administration:* WT mice were treated daily with 0.5 µg rmTPO (Peprotech) or PBS i.v. for 3 days (*Kirito et al., 2002*). Bones were harvested 7 days later. *Immune thrombocytopenia:* WT mice were treated i.v. with 5 µg anti-CD41 (clone MWReg30) or isotype control (Rat IgG1 clone RTK2071) (*Hitchcock et al., 2008*). Bones were harvested 2 days later. Circulating CD41+ platelet were quantified by flow cytometry using 1 µm counting beads (Polysciences, Inc).

### Statistics

Statistical significance in emperipolesis between two conditions was determined using the Chi-square test. Number of MKs counted per sample is reported in Figure legends. To compare emperipolesis in 2 groups of mice we used the Two-tailed Mann-Whitney test. In vitro pro-platelet production (IncuCyte experiment) and in vivo platelet production over time were analyzed with the two-way analysis of variance (ANOVA). All statistical analysis were done using Prism software, $p < 0.05$, $p < 0.01$ $p < 0.001$.

## Acknowledgements

This work was supported by the Arthritis National Research Foundation (to PC); NIH awards 5F32HL118865, K01DK111515, and American Heart Association 16SDG29090007 (to KRM); NIH award R01H168130 (to JEI); and NIH awards R21AR062328, R01AR065538, and P30AR070253, the Cogan Family Foundation, and the Fundación Bechara (to PAN). KRM is an American Society of Hematology Scholar.

## Additional information

### Competing interests

Joseph E Italiano: Is a founder of and has financial interest in Platelet BioGenesis, a company that aims to produce donor-independent human platelets from human induced pluripotent stem cells at scale. JEI's interests were reviewed and are managed by the Brigham and Women's Hospital and Partners HealthCare, in accordance with their conflict-of-interest policies. The other authors declare that no competing interests exist.

## Funding

| Funder | Grant reference number | Author |
| --- | --- | --- |
| Arthritis National Research Foundation | | Pierre Cunin |
| National Institutes of Health | 5F32HL118865 | Kellie R Machlus |
| National Institutes of Health | K01DK111515 | Kellie R Machlus |
| American Heart Association | 16SDG29090007 | Kellie R Machlus |
| American Society of Hematology | Scholar Award | Kellie R Machlus |
| National Institutes of Health | R01H168130 | Joseph E Italiano |
| National Institutes of Health | R21AR062328 | Peter A Nigrovic |
| National Institutes of Health | R01AR065538 | Peter A Nigrovic |
| National Institutes of Health | P30AR070253 | Peter A Nigrovic |
| Cogan Family Foundation | | Peter A Nigrovic |
| Fundación Bechara | | Peter A Nigrovic |

The funders had no role in study design, data collection and interpretation, or the decision to submit the work for publication.

## Author contributions

Pierre Cunin, Conceptualization, Data curation, Formal analysis, Investigation, Methodology, Writing—original draft, Writing—review and editing; Rim Bouslama, Conceptualization, Formal analysis, Investigation, Methodology; Kellie R Machlus, Conceptualization, Resources, Software, Formal analysis, Investigation; Marta Martínez-Bonet, Formal analysis, Investigation, Methodology; Pui Y Lee, Conceptualization, Data curation, Methodology; Alexandra Wactor, Formal analysis, Investigation; Nathan Nelson-Maney, Allyn Morris, Data curation, Formal analysis; Li Guo, Andrew Weyrich, Resources, Validation; Martha Sola-Visner, Resources; Eric Boilard, Joseph E Italiano, Resources, Formal analysis; Peter A Nigrovic, Conceptualization, Resources, Data curation, Formal analysis, Supervision, Funding acquisition, Investigation, Methodology, Writing—original draft, Writing—review and editing

## Author ORCIDs

Pierre Cunin (iD) http://orcid.org/0000-0001-8550-2306
Rim Bouslama (iD) https://orcid.org/0000-0003-2767-6759
Peter A Nigrovic (iD) http://orcid.org/0000-0002-2126-3702

## Ethics

Human subjects: Human marrow was obtained with informed consent under protocol IRB-P00005723 approved by the Institutional Review Board of Boston Children's Hospital.
Animal experimentation: This study was performed in strict accordance with the recommendations in the Guide for the Care and Use of Laboratory Animals of the National Institutes of Health. All animals were handled according to approved institutional animal care and use committee (IACUC) protocols (Dana-Farber Cancer Institute #03-028 and Brigham and Women's Hospital #2016N0005350).

## Decision letter and Author response

Decision letter https://doi.org/10.7554/eLife.44031.025
Author response https://doi.org/10.7554/eLife.44031.026

# Additional files

## Supplementary files

- Transparent reporting form

DOI: https://doi.org/10.7554/eLife.44031.023

## Data availability

All data generated or analyzed during this study are included in the manuscript and supporting files.

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
