## [Decision Letter]

Thank you for submitting your article "Megakaryocyte emperipolesis mediates membrane transfer from intracytoplasmic neutrophils to platelets" for consideration by *eLife*. Your article has been reviewed by three peer reviewers, including Reinhard Fässler as the Reviewing Editor and Reviewer #1, and the evaluation has been overseen by Ivan Dikic as the Senior Editor. The following individual involved in review of your submission has agreed to reveal their identity: Markus Sperandio (Reviewer #2).

The reviewers have discussed the reviews with one another and the Reviewing Editor has drafted this decision to help you prepare a revised submission.

Summary:

The manuscript reports on the mechanism and significance of emperipolesis, a phenomenon by which megakaryocytes (MKs) engulf neutrophils or, to a lesser extent, other hematopoietic cells. The authors established in vitro cultures, with which they define a series of events demonstrating that neutrophils are taken up by MKs in membrane-bound vesicles, are then released into the cytoplasm where they form a membrane continuity with the MK demarcation membrane system, which leads to membrane transfer from neutrophils to MK-derived platelets and to an acceleration of platelet production (shown in inflammation models, LPS treatment and IgG-mediated thrombocytopenia). Furthermore, the authors also demonstrate that there is a reciprocal membrane transfer from MKs to engulfed neutrophils.

Essential revisions:

1) It is not clear from the results whether emperipolesis affects the function(s) of platelets and/or neutrophils. This should be addressed in studying functional aspects of platelets (activation, aggregation, release of cytokines, etc.) from MK cells without neutrophils co-culture or from MK cells co-cultured with neutrophils. Furthermore, we propose to test for neutrophil proteins on platelets by FACS instead of the staining method used in the paper.

2) The cytochalasin D and latrunculin experiments should be better controlled: a positive control (that both drugs effectively disrupt F-actin), quantified by microscopy or FACS, and inactive structural analogs as negative controls.

3) The authors should demonstrate that the neutrophil membrane transfer does not result in an inside-out orientation, which would expose, among other molecules, phospatidylserine and potentially mark the cells for apoptotic uptake by macrophages.

---

## [Author Response]

Essential revisions:1) It is not clear from the results whether emperipolesis affects the function(s) of platelets and/or neutrophils. This should be addressed in studying functional aspects of platelets (activation, aggregation, release of cytokines, etc.) from MK cells without neutrophils co-culture or from MK cells co-cultured with neutrophils. Furthermore, we propose to test for neutrophil proteins on platelets by FACS instead of the staining method used in the paper.

The principal findings of our paper are that it demonstrates, for the first time, two common yet previously unrecognized processes in cell biology:

1) The passage of a mammalian cell through the cytoplasm of another cell from the same organism, in a manner driven actively by both lineages.

2) The transfer of membrane from one lineage to another from an intracellular location, a topologically unique process that explains why neutrophils must assume cytoplasmic residence.

The reviewers raise key questions of the impact of this interaction for the function of platelets and neutrophils. These are important questions, raised also in our Discussion, and we anticipate that they will occupy the lab for better part of the next decade. We agree that the present manuscript will be strengthened by data showing that there is likely to be such impact.

Accordingly, we have now added data reporting the phenotype of hybrid platelets generated in our chimeric WT/mTmG mouse. We show that these platelets express normal levels of the activation marker CD62P (new Figure 5—figure supplement 1H) and show a marginal band of β1-tubulin coil (new Figure 5—figure supplement 1I) excluding an abnormal activation phenotype (Moskalensky et al., 2018; Sadoul, 2015). However they express substantially higher levels of phosphatidylserine, as reflected in Annexin V staining (new Figure 5F and G, control of Annexin V staining specificity in new Figure 5—figure supplement 1G). Surface phosphatidylserine creates a scaffold for clotting factors and is a hallmark of pro-coagulant platelets (Heemskerk et al., 2013; Nagata et al., 2016). This result thereby supports the possibility that emperipolesis-derived platelets are functionally distinct, and potentially pro-thrombogenic, motivating future studies in this direction.

Studies of platelet function, as outlined by the reviewers, will require development of a whole new set of experimental systems. In particular, the WT/mTmG hybrid platelets are identified using a permeabilization strategy to recognize the Tomato stain on the inner membrane leaflet, and so are not suitable for functional work. We are currently experimenting with novel platelet bioreactor systems that will enable generation of platelets in quantities sufficient for studies of activation to diverse ligands, aggregation, and cytokine release. These studies are very important but, we suggest, not essential to the impact of the present paper which focuses on emperipolesis as a novel cell biological phenomenon.

We added the findings related to hybrid platelets phenotype in the Results section as follows:

“Of note, we observed an important fraction of the “red-intermediate” platelets in the mT/mG / WT chimeric mice expressing phosphatidylserine (Figure 5F and G; control for Annexin V staining in Figure 5—figure supplement 1G), while we observe normal levels of CD62P and a marginal band of β1-tubulin by microscopy (Figure 5—figure supplement 1H-I), excluding an abnormal activation phenotype (Moskalensky et al., 2018; Sadoul, 2015). […] This result suggests the possibility that emperipolesis-derived platelets could be functionally distinct, potentially including enhanced thrombogenic capacity.”

We modified the Figure 5 and Figure 5—figure supplement 1 legends accordingly:

Figure 5: “F. Annexin V MFI in WT, mT/mG, and chimeric platelets (control for binding specificity in Figure 5—figure supplement 1G). G. Frequency of Annexin V positive events among WT, mTmG and chimeric platelets. F-G. Representative of 3 experiments.”

Figure 5—figure supplement 1: “E. MKs stained with Cell Tracker Deep Red and co-cultured with surface-biotinylated BM cells were injected i.v. After 1 hour, circulating platelets are stained with Dylight 488-streptavidin. […] I. Platelets from chimeric mice were sorted basted on tdTomato expression and stained with anti-CD41 (green) an anti-β1 tubulin (blue). Representative of 2 experiments. Scale bars represents 1µm.”

We have to date observed no evidence by flow cytometry for neutrophil surface proteins on platelets generated via EP, as determined in platelets generated from MKs engrafted into mice after undergoing this process (not shown). This is in fact the expected result because platelets are generally negative for markers such as CD18, Ly6G, and CD45. Further, studies of emperipolesis of surface-biotintylated bone marrow cells, not included in the original submission but added here in revision, showed that neutrophil surface proteins emerge on MKs but not platelets as determined by streptavidin staining (new Figure 5—figure supplement 1E). Note that these studies do not exclude the possibility that some proteins not studied specifically may still transfer from neutrophils to platelets, and are planning proteomic studies that will specifically address this question.

We report the absence of neutrophil surface proteins on emerging platelets in the Results section as follows:

“We similarly investigated transfer of intracellular or surface protein. Marrow cells were stained with the intracellular protein stain CellTrace Violet and then co-cultured with CMFDA+ MKs. […] Moreover, platelets were negative for neutrophil surface proteins Ly6G, CD11b, CD18 and CD88 (not shown), although we cannot exclude the possibility that other surface proteins not directly assessed may still transfer from marrow cells to platelets in quantities too modest to be detected by bulk biotin-streptavidin staining.”

2) The cytochalasinD and latrunculin experiments should be better controlled: a positive control (that both drugs effectively disrupt F-actin), quantified by microscopy or FACS, and inactive structural analogs as negative controls.

Cytochalasin D (CD) and latrunculin A (LA) are commonly-used reagents that alter the state of actin polymerization by different mechanisms. LA binds actin monomers and prevents them from polymerizing, while also accelerating actin depolymerization (Fujiwara et al., 2018a; Fujiwara et al., 2018b). CD binds actin filaments and blocks their elongation (Brieher, 2013; Cooper, 1987). Both products are cell-permeable and can be used on living cells, and their potential to disrupt the actin cytoskeleton has been established for decades (Cooper, 1987; Lin et al., 1980). While we agree that caution is always appropriate when applying chemical inhibitors, we highlight that the original submission had already featured such an experiment. Specifically, Figure 3—figure supplement 1A showed the efficiency of LA and CD on cytoskeleton disruption using labelled phalloidin, which binds F-actin. This experiment strongly supports the well-established use of these compounds in the way that we have employed them here. Regrettably, we are unaware of inactive structural analogs for LA and CD, though we do not consider this necessarily a limitation of our experimental design given the negative and positive controls already incorporated in our experiments.

In Figure 3—figure supplement 1A, we had chosen to confirm LA and CD efficiency by microscopy using phalloidin because flow cytometry provides information only on the overall quantity of F-actin, which may fail to reflect the true extent of cytoskeleton impairment. For example, cell treatment with CD induces an increase in F-actin signal because CD inhibits F-actin polymerization at a lower rate than it inhibits F-actin depolymerization (Rao et al., 1992; Shoji et al., 2012). However, to address the reviewer request, we repeated the experiment in Figure 3—figure supplement 1A but used flow cytometry to monitor F-actin. As expected, we found that F-actin signal is reduced with LA, while CD induces an increase of F-actin signal (Author response image 1). Our preference is to include this information for the reviewers only, rather than in the manuscript itself, because we believe that the existing Figure 3—figure supplement 1A better reflects the biologic impact of these inhibitors.

**Author response image 1. respfig1:** Marrow cells were treated with 1µM latrunculin A, cytochalasin D, or DMSO. Inhibitors were either left in culture (**A**), or thoroughly washed after 2 hours (**B**). After overnight incubation, cells were permeabilized and stained with labelled phalloidin. Left: Phalloidin MFI on Ly6G+ neutrophils, representative of 3 experiments. Right: Phalloidin MFI expressed as a percentage of control (DMSO condition), n=3.

We emphasize further that our evidence for the importance of actin in emperipolesis is not based only on the use of chemical inhibitors. The observation of neutrophil polarization during entry into MK (e.g. Video 2), and the formation of a transmembrane cup on MK surface (e.g. Videos 2 and 5) had strongly implicated both MK actin and neutrophil actin in emperipolesis. Further, confocal microscopy demonstrated particularly prominent actin polymerization at the MK surface where neutrophils are attached, as well as around “emperisomes” containing neutrophils (Figure 3C) where F-actin colocalized strikingly with the CD41 marker that delineates this vacuole (Figure 3C and Figure 3—figure supplement 1C). Taken together, we suggest that these data conclusively establish the role of actin in emperipolesis.

3) The authors should demonstrate that the neutrophil membrane transfer does not result in an inside-out orientation, which would expose, among other molecules, phospatidylserine and potentially mark the cells for apoptotic uptake by macrophages.

We are grateful for this intriguing question. To our estimation, the topology of emperipolesis would not particularly favor inversion of the neutrophil membrane during either cytoplasmic entry or membrane fusion with the demarcation membrane zone. Similarly, we observed no visible alterations in neutrophils following their exit from MKs. In particular, these neutrophils retain a normal nucleus shape and continue to diapedese freely among the other bone marrow cells in the culture. Thus none of our data to date suggest a role of emperipolesis in neutrophil clearance, though our data also do not formally exclude this possibility, and we remain very intrigued by the ways that trans-MK passage alters neutrophil function and fate.

We therefore sought to address the question experimentally. We labeled marrow cell with the membrane dye CellVue maroon to track membrane transferred during emperipolesis. Bone marrow cells were then co-cultured with MKs in the presence of FITC-labelled lactadherin (also known as MFGE8), an opsonin that binds specifically to phosphatidylserine in a Ca^2+^-independent manner (Hanayama et al., 2002). While we could again demonstrate membrane transfer to the MK DMS, we failed to observe lactadherin binding to these membranes, confirming that the transfer of membrane does not result in phosphatidylserine exteriorization (Author response image 2 and new Figure 4—figure supplement 1E). Moreover, neutrophils do not overexpress phosphatidylserine when co-cultured with MKs, as evidenced by the absence of Annexin V signal on their surface by flow cytometry (New Figure 4—figure supplement 1F). Thus, while we cannot yet exclude that MK passage labels neutrophils for accelerated clearance via phagocytes, we do not observe any evidence cells are so labeled via enhanced phosphatidyserine exposure.

**Author response image 2. respfig2:** Bone marrowcells were stained with CellVue Maroon prior to co-culture with MKs, as in Figure 4B. 2µM of labelled lactadherin (LD, also known as MFGE8) was added in the culture. Confocal microscopy show a transfer of CellVue maroon (white) in the MK DMS (green). The absence of LD binding (blue) demonstrates that phosphatidylserine is not externalized during the process. LD positive cell is shown in Figure 4—figure supplement 1E. Representative of 2 experiments.

Text was added to the Results section as follows:

“Of note, membrane transfer during emperipolesis was not associated with phosphatidylserine externalization onto the neutrophil surface (Figure 4—figure supplement 1E-F).”

We modified the Figure 4—figure supplement 1 legend accordingly:

“E.Bone marrowcells were stained with CellVue Maroon prior to co-culture with MKs, as in Figure 4B. […] Representative of 2 experiments.”